# Changes in Saliva Analytes Correlate with Horses’ Behavioural Reactions to An Acute Stressor: A Pilot Study

**DOI:** 10.3390/ani9110993

**Published:** 2019-11-18

**Authors:** María D. Contreras-Aguilar, Séverine Henry, Caroline Coste, Fernando Tecles, Damián Escribano, Jose J. Cerón, Martine Hausberger

**Affiliations:** 1Interdisciplinary Laboratory of Clinical Analysis (Interlab-UMU), Campus of Excellence Mare Nostrum, 30100 Murcia, Spain; 2Université de Rennes, Université de Normandie, CNRS, Laboratoire Ethologie et humaine - UMR CNRS 6552, Station Biologique, 35380 Paimpont, France; 3Department of Animal Production, Campus of Excellence Mare Nostrum, University of Murcia, 30100 Espinardo, Murcia, Spain; 4Université de Rennes, Université de Normandie, CNRS, Laboratoire Ethologie et humaine - UMR CNRS 6552, Campus de Beaulieu, Bat.25, 263 avenue du général Leclerc, 35042 Rennes CEDEX, France

**Keywords:** acute stress, biomarkers, horse, stress behaviour, saliva

## Abstract

**Simple Summary:**

Emotionality is an individual characteristic defined as the propensity to respond to stress-inducing stimuli such as fear-inducing objects or social separation. Evaluation of emotionality in horses is important as it may impact their learning performance. Although emotionality is usually assessed by measuring behavioural patterns, biomarkers could provide additional information about stress response, especially with respect to its temporal dynamics. In this study, behavioural responses were measured as well as a panel of salivary biomarkers related to stress, including salivary alpha-amylase, lipase, total esterase, butyrylcholinesterase, adenosine deaminase, and cortisol, in riding horses after acute experimental stress (the sudden opening of an umbrella). We found significant changes in most of the salivary biomarkers evaluated after the induced stress, where increases in butyrylcholinesterase were more closely related to behavioural responses to acute stress and low salivary alpha-amylase values were more closely related to quietness behaviours. Therefore, this preliminary research provides information about the relationship between behaviour patterns and biomarkers of stress in saliva in horses, and opens the possibility of wider use of selected biomarkers in saliva, such as butyrylcholinesterase or alpha-amylase, for the evaluation of acute stress in horses.

**Abstract:**

Acute stress induces an array of behavioural reactions in horses that vary between individuals. Attempts to relate behavioural patterns and physiological responses have not always given clear-cut results. Here, we measured the changes in a panel of salivary components: salivary alpha-amylase (sAA), lipase, total esterase (TEA), butyrylcholinesterase (BChE), adenosine deaminase (ADA), and cortisol, and their potential link with horses’ behaviours after acute stress. Saliva samples were collected in nine riding horses subjected to a test consisting of opening an umbrella. Saliva sampling was obtained at a basal time point in the stall (T1), in the test indoor arena (T2), at a time of stress (T3), and 30 min (T4) and 60 min (T5) later. The horses’ behaviour was recorded at T3 for 1 min. sAA, lipase, TEA, and BChE showed significant changes along time, increasing at T3 for BChE, and decreasing at T4 for sAA and BChE. Butyrylcholinesterase appeared to be the most reliable predictor of behavioural responses, as it correlated with the index of emotionality, of laterality, and the occurrence of alarm signals, while sAA decreased when horses expressed quieter behaviours. These first results bring promising lines for novel, more precise physiological markers of acute stress in horses that can bridge the gap between behaviour and physiology.

## 1. Introduction

The evaluation of horses’ emotionality is important in riding horses, as strong emotional reactions can be associated with impaired learning performances (e.g., in spatial learning or instrumental tasks) [1,2,3]. Emotionality is generally assessed by measuring the behaviours expressed by horses in standardized experimental tests, such as the surprise or the bridge test which assess “fearfulness”, the arena test related to “gregariousness”, or the novel object test correlated with “nervousness” [4]. Some authors use the behaviours expressed in these tests to build an index based on both behavioural patterns and their frequencies of occurrence, that allows ranking of the animals tested in the same conditions [5,6]. However, measures of the behavioural components do not give a complete information about the physiological mechanism of the stress response. Therefore, it could be useful to develop further physiological measures as some authors have suggested [7].

Saliva is easily collected through non-invasive and non-stressful procedures [8,9] and until now, most studies related to acute stress measurements in horses have focused on salivary cortisol [10,11,12]. This informs about the free cortisol concentration, which is the active part of the hormone [13], to therefore assess the hypothalamic–pituitary–adrenal (HPA) axis [14,15]. This measure has proved useful to measure the immediate response of foals to the acute stress of maternal separation on artificial weaning in particular [16]. However, these studies do not evaluate the possible correlations between cortisol and the precise behavioural changes in response to an acute stressor. In addition, there is a lack of knowledge about if other salivary analytes that have been reported to change after a stress in other species could vary also in horses, and their possible relationship with behavioural changes. For example, salivary alpha-amylase (sAA), considered an autonomic nervous system (ANS) marker as it is released into oral cavity after activation of beta-adrenergic receptors by intracellular noradrenaline [17], is in high levels in horses with acute abdominal disease (AAD) [18,19] or in sheep after two acute fear-inducing stressors [20]. In addition, other salivary analytes such as total esterase (TEA), butyrylcholinesterase (BChE), and lipase increased at different levels in pigs and sheep after acute stress situations [20,21,22], demonstrating a relationship with sAA [20]. Finally, adenosine deaminase (ADA) is an enzyme considered to be a biomarker of cell-mediated immunity [23], and it plays an important role in detoxification [24], but it also increases in rats’ blood and in sheep’s saliva after acute stress [20,25], suggesting a link between salivary ADA and stress axis.

In this study, we hypothesized that sAA, lipase, TEA, BChE, ADA, and cortisol levels in saliva could be influenced by the application of an acute stressor and would correlate with the behavioural stress responses in horses. Therefore, the aims of this study were to evaluate the possible changes in those salivary analytes and their potential correlation with horses’ behaviour after an experimental acute stress consisting of a fearfulness test based on suddenness: opening an umbrella in front of riding horses just after releasing them in an indoor familiar covered arena.

## 2. Materials and Methods 

Nine horses of different ages (5 to 22 years old, mean + standard error = 13 + 4.8), sex (4 geldings and 5 mares), and of varied breeds (French Saddlebreds, mixed breeds, and unregistered horses) from a riding school in Brittany (France) were involved in this study. Housing conditions were single 3 m × 3 m boxes in a barn with door openings and grids in the wall allowing visual contact with conspecifics, although horses spent an additional 6 h per day in paddocks, where horses were in variable groups (from 2 to 11 individuals). Horses were fed industrial pellets twice a day (08:00 h and 19:00 h) and hay was provided ad libitum once a day (09:00 h). Water was available ad libitum. Horses were working in riding lessons with typical English riding style for 4–12 h per week under supervision of a riding teacher [26,27].

Saliva was obtained through a device previously reported by Henry et al. [16]. A natural fibre gauze was introduced in a plastic tube with a hole cut longitudinally in it, and suspender clips were attached to each end of the tube (Figure 1) and connected the device to the halter using swivel-head clips to place it inside the horse’s mouth. The horse was allowed to chew on the device during 1 min. The gauze was then removed from the tube, placed into a collection device (Salivette, Sarstedt, Aktiengesellschaft & Co, Nümbrecht, Germany) and stored on ice. At the laboratory, salivettes were centrifuged at 3000 g for 10 min at 4 °C, and saliva was then transferred into 1.5-mL Eppendorf tubes and stored at −80 °C until analysis (less than one month). All procedures involving animals were in accordance with the ethical standards of the Bioethical Commission of Murcia University (CEEA 288/ 2017).

The experiment took place at a basal time point (T1) in box stalls. Then, each horse was halter-led and taken to the centre of the indoor riding arena (40 m × 20 m) in front of a person (S.H., 3 m distance) with a closed dark umbrella (Figure 2); then, a second sample (T2) was collected. The horses were released, the umbrella was suddenly opened, and T3 sample (stress time) was obtained while the device was inside horses’ mouth. After one minute, horses were caught and led again to their individual boxes. Finally, once the horses were in their box stalls, two additional saliva samples were taken 30 min (T4) and 60 min (T5) after T3.

The experimental procedure was performed in calm weather, and all horses were tested on the same day from 09:00 h to 12:30 h. The indoor riding arena was behind the individual box stalls, and horses could not see what happened in the inside of the arena (Figure 2). The same two experimenters (M.D.C.A. and M.H.) each handled those horses for which they had performed the saliva samplings (5 each).

Horses’ behaviour was video-recorded (JVC EverioR GZ-RX645 BE, JVCKENWOOD, Japan; positioned behind the person with the umbrella, S.H.) during T3 (Figure 2) for later analysis. The following behavioural patterns were recorded: (1) standing quietly immobile, (2) a slow/exploratory walk (the horse walks slowly with its neck held horizontally or below, ready to stop and to sniff the ground; a characteristic walk of a quiet horse in a calm situation), (3) a sustained walk (the horse walks energetically, and looks ahead or around), (4) a trot or gallop (5), passage (sustained trot, with legs raised higher), (6) vigilance (the horse stands still, holding its neck high (>45°) with intently oriented head and ears), (7) tail posture (the tail can hang down or be raised, the fleshy portion of the tail then is held almost or completely upright, with the long hairs of the tail stream making a showy display, (6) alarm (the horse stands still but with the neck higher than in vigilance, with intently oriented head and ears and farther from the stressor), (7) alarm with acoustic signals such as snoring (a very short raspy inhalation sound produced in a low alert context) and blowing (a short very intense non-pulsed exhalation through the nostrils), (8) looking at the umbrella with both eyes or with the right eye (RE) or left eye (LE), (9) and ear orientation (axial ear when perpendicular to the head–rump axis, backward ear when the tip of the ear is towards the back at more than 30° from perpendicular, forward ear when the tip of the ear is towards the front at an angle of more than 30° from perpendicular, and mixing ear when one ear is in a forward and the other is in a backward orientation) [1,5,7,27]. The horses’ behaviour was then analysed using scan sampling (one scan every 2 s) during 1 min (30 scans/min), except for alarm with acoustic signals which were noted every time they occurred since they are considered short events [6,28].

The salivary alpha-amylase (sAA) activity (IU/L) was measured using a colorimetric commercial kit (Alpha-Amylase, Beckman Coulter Inc., Fullerton, California, USA), as previously reported [29] and validated [30] in horses. The lipase activity (IU/L) in the saliva was measured using a commercially available method (Lipase, Beckman Coulter Inc.) based on the ability of lipase to hydrolyze the heavy-chain esterified fatty acid 1,2-diglyceride. The total adenosine deaminase (ADA) activity (IU/L) was measured using a spectrophotometric automated method (Adenosine Deaminase Assay Kit, Diazyme Laboratories, Poway, CA, USA) based on the enzymatic deamination of adenosine to inosine, which is specific for this enzyme [31]. Total esterase (TEA, IU/L) was analysed as previously reported [21] using 4-nitrophenyl acetate (Sigma-Aldrich Co.) as a substrate. Butyrylcholinesterase (BChE, nmol/mL/min) was analysed as previously described [22] using the chromophore 5,5’-dithiobis-2-nitrobenzoic acid (Sigma-Aldrich Co., St Louis, MO, USA) and the substrate butyrylthiocholine iodide (Sigma-Aldrich Co.) which is specific to this enzyme [21]. All of these analytes have been previously validated in horses [18,30] using an automated biochemical analyser (Olympus Diagnostica GmbH AU 400, Beckman Coulter, Ennis, Ireland). Salivary cortisol (μg/dL) was analysed using a chemiluminescent immunoassay system (Immulite 1000, Siemens Healthcare Diagnostic, Deerfields, Illinois, USA). This was previously validated in saliva for horses [30] and pigs [32].

Frequencies of occurrence of behavioural patterns were calculated [28] and in order to rank the behavioural reactions of the experimental horses at T3, an index of emotionality previously validated based on six behavioural patterns was used [4,5,6]. Values were attributed to the behavioural patterns according to their degree of specificity and level of arousal. These values were exploration/slow walk = 1, sustained walk = 2, trot or gallop = 3, vigilance = 4, alert = 5, and prancing, snoring/blowing or tail raised = 6. These values were multiplied by the number of times the corresponding pattern was observed according to the following formula: index of emotionality = ((times that exploration/slow walk appears) × 1) + ((times that sustained walk appears) × 2) + ((times that trot or gallop appears) × 3) + ((times that vigilance appears) × 4) + ((times that alertness appears) × 5) + ((prancing, snoring/blowing or tail raised) × 6). It must be stressed that these values give only a rank indication and do not represent real data. Thus, a horse with an index twice as high as that of another horse was not necessarily twice as reactive. Additionally, the index of laterality was calculated based on the following formula: index of laterality = ((right eye) – (left eye))/((right eye) + (left eye)) [33], which ranges from –1.0 to 1.0 (reflecting a preferential use of the right eye, i.e., positive emotions) [34,35].

Data were checked for normality using the Shapiro–Wilk normality test. sAA, lipase, ADA, BChE, and cortisol in saliva, and frequency of occurrence of standing quietly immobile, slow walk, sustained walk, trot and gallop, passage, tail raised, alarm, snoring/blowing, sniffing at the ground, and axial and backward ear orientation, showed a non-normal distribution. Salivary analytes that showed non-normal distribution were inversely transformed before the analysis one-way ANOVA of repeated measures followed by Tukey’s multiple comparison test to assess whether or not the different salivary analytes had changed at different times after the stress stimuli. Additionally, a post hoc analysis by a stand-alone power program for statistical testing (G-Power) [36] was employed using the means and standard deviations of the salivary analytes to guarantee that the significance level (α = 5%, *p* < 0.05) and the power required (1 − β ≥ 80%) were correctly obtained with the number of horses evaluated. 

Spearman’s or Pearson’s *r* were calculated between the salivary analytes at the stress time (T3) and the behaviour parameters depending on whether the data were non-normally or normally distributed, in order to know whether there was correlation between them. An *r* value of less than 0.30 was considered as a negligible correlation, following the Rule of Thumb [37]. A *p*-value less than 0.05 was considered as being statistically significant. Data analyses were performed using spread-sheet software (Excel 2000; Microsoft Corporation, Redmond, WA, USA) and Graph Pad Software Inc (GraphPad Prism, version 6.0c for Mac; Graph Pad Software Inc., San Diego, CA, USA).

## 3. Results

### 3.1. Salivary Analytes Results

Salivary analytes results are shown in the Figure 3. Salivary alpha-amylase (sAA, F_2.52,20.20_ = 3.25, *p* = 0.050), lipase (F_2.59,20.68_ = 7.83, *p* = 0.002), total esterase (TEA, F_2.69,21.55_ = 5.15, *p* = 0.009) and butyrylcholinesterase (BChE, F_2.23,17.84_ = 7.30, *p* = 0.004) showed significant changes between the sampling times, with an increase at T3 with respect to T1 in BChE (*p* = 0.038) and TEA (*p* = 0.026), and a decrease at T4 and T5 in sAA (*p* = 0.035 and *p* = 0.032, respectively) and BChE (*p* = 0.050 and *p* = 0.002, respectively) with respect to T3. Meanwhile, increases at T2 (*p* = 0.009), T3 (*p* = 0.004), T4 (*p* = 0.037), and T5 (*p* = 0.014) from T1 were observed in lipase. Adenosine deaminase (ADA) did not show significant changes over time (F_2.39,19.14_ = 2.77, *p* = 0.079). Although the ANOVA of repeated measures did not point changes between the sampling times in salivary cortisol (F_2.56,20.49_ = 1.90, *p* = 0.167), the multiple comparison test showed a decrease from T3 to T4 that was in the limit of significance (*p* = 0.050).

The post hoc power analysis for the statistical analysis performed on the analytes showed a power (1−β) for sAA of 81%, lipase of 99%, TEA of 98%, BChE of 98%, ADA of 71%, and cortisol of 66% in the population evaluated for the present study (*n* = 9).

### 3.2. Correlations Between Analytes and Behaviour Patterns

Table 1 shows the frequencies of occurrence of the different behavioural patterns, the index of emotionality and the index of laterality obtained for each horse. Table 2 shows the correlation matrix between the salivary analytes and the behavioural pattern. BChE was correlated with the index of emotionality (r = 0.68, *p* = 0.049) and with the frequency of occurrence of “glances to umbrella with both eyes” (r = 0.69, *p* = 0.044) (Figure 4). Negative correlations were observed between some analytes and “quiet behaviours” such as “sniffing at the ground” (sAA: r = −0.65, *p* = 0.043) and “standing quietly immobile” (BChE: r = −0.69, *p* = 0.041) (Figure 4), and the index of laterality (ADA, BChE: r = −0.68, *p* = 0.041). In addition, positive correlations between BChE levels and the frequency of occurrence of alarm acoustic signals (snoring/blowing) (r = 0.69, *p* = 0.046), and between cortisol and BChE levels and frequency of occurrence of “glances to the umbrella with the left eye” (r = 0.71, *p* = 0.039; r = 0.76, *p* = 0.025; respectively) were observed (Figure 4).

## 4. Discussion

To the authors’ knowledge, this is the first study showing a relationship between stress-related salivary biomarkers and behavioural reactions to acute stress in horses.

Most analytes measured in saliva showed changes of different magnitude and temporal sequence after the induction of the stress in horses. Salivary alpha-amylase (sAA) and lipase increased after stress induction, which seems to indicate an activation of the autonomic nervous system (ANS) [17,38]. Previous reports have demonstrated increases in these analytes after acute stress in sheep [20] and pigs [21,39]. In our study in horses, none of these two analytes showed any correlation between them, with a different pattern of changes and sAA returning to normal values sooner than lipase (30 vs longer than 60 min). On the other hand, salivary cortisol, which is traditionally considered as a marker of the HPA axis, increased at the stress time, returning to basal levels 30 min later. This increase occurs sooner than previously reported, since according to these earlier studies, salivary cortisol in horses can increase only 30 min after adrenocorticotropic hormone (ACTH) stimulation [15,40]. A possible hypothesis about the fast increase in the measured salivary cortisol in our study could be due to the contraction of the salivary glands’ myoepithelial cells as results of ANS activation [41], which could affect cortisol concentration in saliva [42]. This may be the reason for the significant correlation found between salivary cortisol and sAA. Finally, total esterase (TEA) and butyrilcholinesterase (BChE) statistically increased at the stress time, with BChE showing a later decrease to basal levels. Both markers have been associated to acute stress situations in other domestic animals, such as in pigs [21,22] or sheep [20], and increases in salivary BChE have been associated with decreased parasympathetic activity in human beings [43], that can appear in stressful situations [44]. Overall, these results indicate that the model of stress used in our study produced changes in most salivary analytes tested of the horses.

BChE correlated with the index of emotionality, calculated by combining the behavioural responses to a fear-inducing stressor in horses. Additionally, the positive correlation of BChE with the frequency of occurrence of “glances to the umbrella with both eyes”, alarm with acoustic signals (snoring/blowing) associated with vigilance/alarm postures [27], and “glances to the umbrella with the left eye”, meaning a predominance of the right hemisphere associated with heightened fear and/or withdrawal in horses [34,35,45], as well the negative correlation with other “quiet behaviours”, would reflect an association between increases of BChE activity and acute stress in horses. Although there is evidence that BChE is directly released after acute stress in other species [20,22,46], further studies are necessary to assess the possible physiological mechanisms relating to increases of BChE in saliva and stress-related behaviour in horses. 

Salivary cortisol correlated highly with the left eye’s glances. However, in a study performed in dogs [47], behavioural measurements did not correlate with salivary cortisol after different types of stressful stimulus, probably due to the high inter-individual variation in stress response reported. Salivary alpha-amylase had a negative moderate correlation with “quiet behaviours” such as sniffing at the ground, which could indicates that sAA is more related with movement, in agreement with the increases in sAA reported in situations of exercise [30]. Although ADA did not change significantly after stress, it correlated negatively with the index of laterality, with the preferential use of the left eye, and positively with BChE. Further studies must be performed to assess the possible link between stress and ADA, since increases in ADA in saliva have been reported in sheep after two different stressful situations [20].

Some limitations should be taken into account in this research. The population selected showed enough statistical power for all analytes with exception of salivary cortisol and adenosine deaminase (ADA), but the sample remained low. Therefore, this study should be considered as a pilot one and further studies involving a larger number of horses would be recommended in order to confirm these preliminary results. In addition, this study has been performed under fixed experimental conditions; therefore, other stressful situations and with another kind of populations should be assessed to confirm the generality of our results. For instance, the population in this study was only composed of mares and geldings, whereas possible differences between sex (stallions vs. geldings vs. mares) or reproductive state could affect basal levels for some salivary biomarkers in horses, such as in salivary cortisol [40,48]. Also, there are differences in stress behaviour pattern according to breed, housing condition, or the type of work performed by the horses [1,5], and the temporal dynamic of behavioural patterns was not evaluated. Finally, the results could have potentially changed if other materials for saliva collection had been used as previously described [49]. 

In conclusion, sAA, lipase, TEA, BChE, and cortisol changed in saliva due to sudden stressful stimuli related to fearfulness in horses. BChE was more closely related to acute stress and low sAA values to quietness behaviours. Although further studies with a larger number of horses must be performed to confirm our preliminary results, this research support the possibility of wider use of saliva as a sample for evaluation of acute stress in horses by the measurement of analytes such as BChE and sAA, that may prove more reliable than classical measures of cortisol only, and open up a new field of research about the possible associations between behaviours and changes in salivary biomarkers in the horse. Since welfare may be impaired by repeated acute stress, it would be worth also extending this research to more chronic stress situations.

## Figures and Tables

**Figure 1 animals-09-00993-f001:**
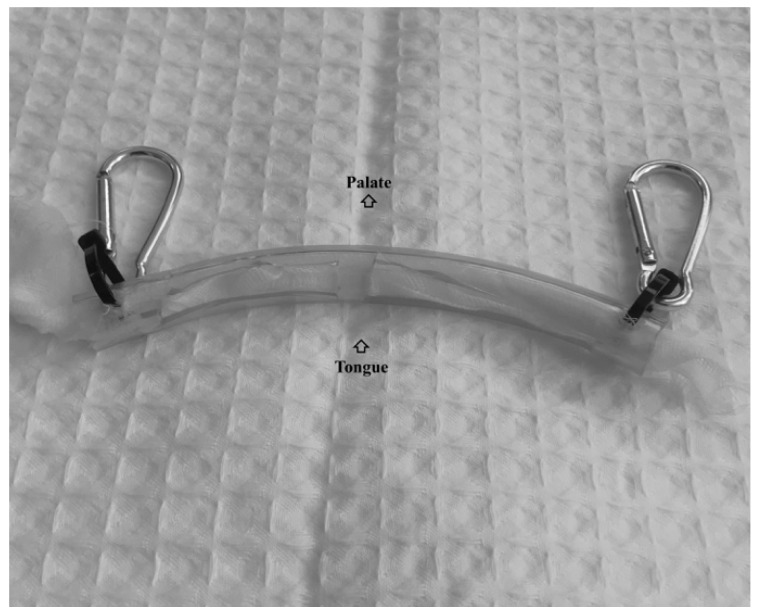
Device employed to collect saliva.

**Figure 2 animals-09-00993-f002:**
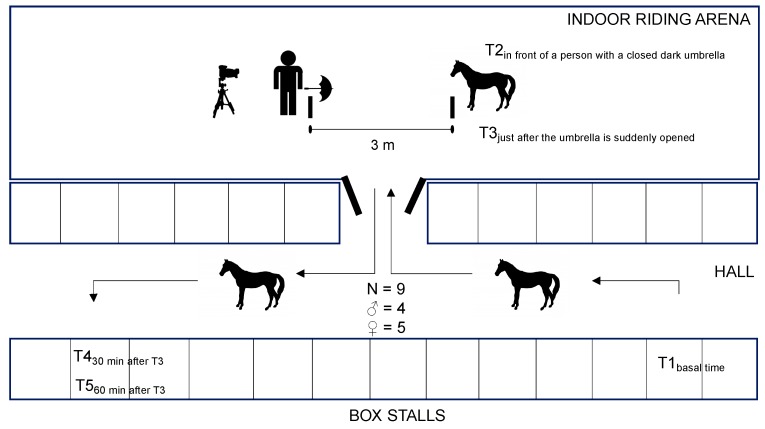
Scheme of the experimental design.

**Figure 3 animals-09-00993-f003:**
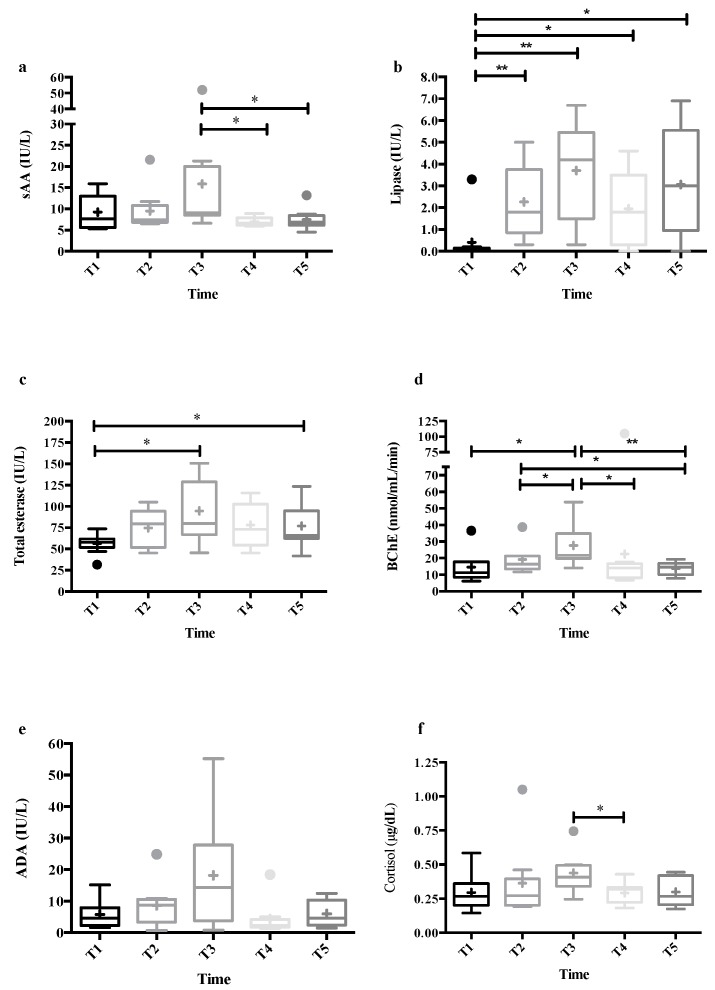
Results of salivary alpha-amylase (sAA) (**a**), lipase (**b**), total esterase (TEA) (**c**), butyrilcholinesterase (BChE), (**d**), adenosine deaminase (ADA) (**e**), and cortisol (**f**) in the saliva of nine horses during a sudden experimental stress. Saliva samples were taken at basal time (T1) in their box stalls, when each horse was led by halter in the middle of an indoor riding arena (T2), just after the umbrella was suddenly opened once the horses were released (T3), and 30 min (T4) and 60 min (T5) after the stressor stimulus in their box stalls. The plots show median (line within box), 25th and 75th percentiles (box), 5th and 95th percentiles (whiskers), and outliers (●). The cross inside the box shows the mean. Asterisk indicates statistically significant difference (* *p* < 0.05, ** *p* < 0.01) between times.

**Figure 4 animals-09-00993-f004:**
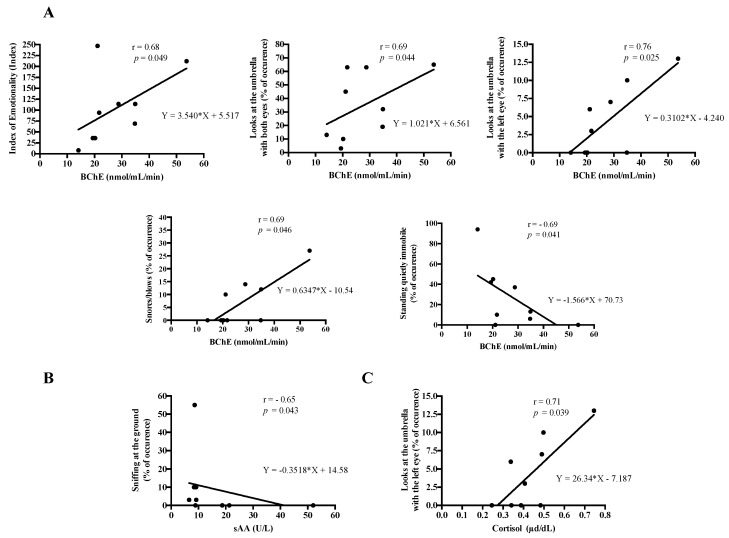
Linear regression of correlation plots from the analytes that showed statistical significance with the frequencies of occurrence of some behavioural patterns. Butyrilcholinesterase (BChE) (**A**), salivary alpha-amylase (sAA) (**B**), and salivary cortisol (**C**). r = Spearman’s r-value.

**Table 1 animals-09-00993-t001:** Frequencies of occurrence of behavioural patterns, the index of emotionality and the index of laterality obtained from nine horses after a sudden experimental stress consisting in opening an umbrella in front of them. Horses’ behaviour was recorded during 1 min by scan sampling (30 scans/min).

	Horse 1	Horse 2	Horse 3	Horse 4	Horse 5	Horse 6	Horse 7	Horse 8	Horse 9	Mean/Median ^1^
**Behavioural patterns, frequencies of occurrence (%)**										
Standing quietly immobile	10	13	94	0	37	45	6	0	42	13 [3.0–43.5]
Slow/exploratory walk	0	0	0	0	0	16	42	3	39	0 [0.0–27.5]
Sniffing at the ground	0	3	10	0	10	55	0	3	0	3 [0.0–10.0]
Sustained walk	26	55	0	26	13	0	0	23	0	13 [0.0–26.0]
Trot	6	3	0	19	7	0	0	13	0	3 [0.0–10.0]
Gallop	0	0	0	0	3	0	0	10	0	0 [0.0–1.5]
Passage	0	3	0	0	0	0	0	6	0	0 [0.0–1.5]
Vigilance	56	26	6	55	43	3	45	29	19	31 (16.2–46.5)
Tail posture	0	3	0	19	0	0	0	26	0	0 [0.0–11.0]
Alarm	0	0	0	0	0	0	0	3	0	0 [0.0–0.0]
Snoring/blowing	0	12	0	27	14	0	0	10	0	0 [0.0–27.0]
Looking at the umbrella	69	55	23	97	73	16	19	77	17	49 (25.5–73.4)
*with both eyes*	63	32	13	65	63	10	19	45	3	31 (13.1–49.9)
*with the right eye*	3	13	10	19	3	6	0	26	14	10 (3.9–17.0)
*with the left eye*	3	10	0	13	7	0	0	6	0	4 (0.5–8.1)
Ear orientation										
*axial ear*	10	10	17	3	4	77	29	3	7	10 [3.5–23.0]
*backward ear*	7	7	31	3	15	6	14	0	39	7 [4.5–23.0]
*forward ear*	77	70	34	77	78	13	50	90	50	60 (40.7–79.1)
*mixing ear*	7	13	17	16	4	3	7	6	4	8 (4.4–12.7)
**Index of emotionality**	94	114	8	212	114	36	69	247	36	103 (36.0–212.0)
**Index of laterality**	0	0.14	1.00	0.20	−0.33	1.00	0	0.60	1.00	0.40 (0.001–0.793)

^1^ Mean (95% confidence interval); Median [interquartile range, 25th–75th percentiles].

**Table 2 animals-09-00993-t002:** Correlation coefficients between salivary analytes (salivary alpha-amylase (sAA), lipase, total esterase (TEA), butyrylcholinesterase (BChE), adenosine deaminase (ADA), and cortisol) and the frequencies of occurrence of behavioural patterns, the index of emotionality, and the index of laterality obtained in nine horses after a sudden experimental stress.

	sAA (IU/L)		Lipase (IU/L)		TEA (IU/L)		BChE (nmol/mL/min)		ADA (IU/L)		Cortisol (μg/dL)	
	r	*p* value	r	*p* value	r	*p* value	r	*p* value	r	*p* value	r	*p* value
sAA (IU/L)			0.15	0.334	0.01	0.987	**0.30**	**0.049**	0.08	0.582	**0.40**	**0.006**
Lipase (IU/L)	0.15	0.334			**0.69**	**<0.001**	**0.36**	**0.014**	0.23	0.126	0.26	0.088
TEA (IU/L)	0.01	0.987	**0.69**	**<0.001**			0.24	0.110	**0.49**	**<0.001**	**0.31**	**0.040**
BChE (nmol/mL/min)	**0.30**	**0.049**	**0.36**	**0.014**	0.24	0.110			**0.44**	**0.002**	**0.43**	**0.004**
ADA (IU/L)	0.08	0.582	0.23	0.126	**0.49**	**<0.001**	**0.44**	0.002			0.22	0.143
Cortisol (μg/dL)	**0.40**	**0.006**	0.26	0.088	**0.31**	**0.040**	**0.43**	**0.004**	0.22	0.143		
Standing quietly immobile (%)	−0.26	0.482	−0.18	0.633	−0.08	0.817	**−0.69**	**0.041**	−0.16	0.665	−0.21	0.571
Slow/exploratory walk (%)	0.07	0.870	0.16	0.680	0.02	0.870	−0.26	0.368	−0.18	0.647	−0.57	0.062
Sniffing at the ground (%)	**−0.65**	**0.043**	0.18	0.651	−0.12	0.761	−0.39	0.236	−0.09	0.717	0.11	0.770
Sustained walk (%)	0.01	0.901	−0.08	0.734	0.08	0.842	0.67	0.06	0.19	0.620	0.59	0.098
Trot (%)	0.16	0.686	−0.12	0.668	−0.18	0.556	0.57	0.116	−0.03	0.844	0.45	0.220
Gallop (%)	−0.16	0.222	0.09	0.583	−0.07	0.306	−0.02	0.389	0.07	0.556	−0.16	0.222
Passage (%)	−0.25	0.111	0.55	0.139	0.48	0.222	0.16	0.694	0.16	0.694	−0.07	0.306
Vigilance (%)	0.38	0.313	−0.25	0.521	−0.22	0.581	0.67	0.06	0.33	0.385	0.32	0.410
Tail posture (%)	0.01	0.663	0.42	0.266	0.30	0.440	0.46	0.226	−0.08	0.492	0.540	0.540
Alarm (%)	0.14	0.222	0.55	0.222	0.14	0.222	−0.13	< 0.001	−0.14	<0.001	**−0.41**	**<0.001**
Snoring/blowing (%)	−0.29	0.310	−0.18	0.476	0.05	0.900	**0.69**	**0.046**	0.13	0.760	0.67	0.057
Looks at the umbrella (%)	0.17	0.678	0.00	0.999	−0.14	0.725	0.53	0.146	0.15	0.708	0.38	0.312
*with both eyes*	0.11	0.784	−0.18	0.620	−0.20	0.599	**0.69**	**0.044**	0.23	0.546	0.60	0.092
*with the right eye*	0.26	0.498	0.25	0.512	0.03	0.935	−0.06	0.868	−0.50	0.164	−0.12	0.751
*with the left eye*	−0.16	0.603	−0.17	0.571	0.09	0.823	**0.76**	**0.025**	0.16	0.686	**0.71**	**0.039**
Ear orientation (%)												
*axial ear*	−0.31	0.398	0.20	0.600	0.36	0.337	−0.30	0.411	0.15	0.698	−0.08	0.816
*backward ear*	−0.02	0.958	−0.36	0.327	−0.21	0.572	−0.42	0.242	0.26	0.498	−0.38	0.304
*forward ear*	0.15	0.697	−0.13	0.715	−0.02	0.950	0.45	0.232	0.18	0.646	0.16	0.684
*mixing ear*	0.11	0.785	0.41	0.272	0.44	0.234	0.29	0.456	0.43	0.249	0.23	0.555
Index of emotionality (index)	−0.03	0.920	0.03	0.940	0.05	0.899	**0.68**	**0.049**	0.12	0.767	0.40	0.281
Index of laterality (index)	0.21	0.581	0.32	0.391	−0.11	0.779	**−0.68**	**0.041**	**−0.68**	**0.041**	−0.45	0.195

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
