# Peer review of "Changes in Saliva Analytes Correlate with Horses’ Behavioural Reactions to An Acute Stressor: A Pilot Study"

_animals, 2019, doi:10.3390/ani9110993_

Round 1

Reviewer 1 Report

Review Manuscript ID: animals-629543

Comments

I believe the manuscript (ID: animals-629543) outlines adequately a prospective study that evaluated evaluate possible changes in salivary analytes and their potential correlation with horses’ behaviour after an experimental acute stress.

The reported study covers an interesting topic that is worthy of study and I believe the manuscript is suited for publication with minor revisions. 

Minor comments:

Introduction

Line 49: you use may twice in the same sentence, consider change to could ((e.g. flight behaviours,) and could be associated)

M&Ms

Line 86: spell out to X +SE to mean and standard error

Line 86: change to “sex”

Line 89: change to “boxes in a barn” (delete housing)

Line 91: Change “kept in unstable groups” to “variable groups”

Line 94: it is not clear why the reader should “see also ref 26 and 27”

Line 95: change “device used by Henry et al., [16]” to “device previously reported by Henry et al., [16]”

Line 96: it would be more explanatory if you provide a picture of the device

Line 105: change to “haltered and taken to the center…”

Line 106: changed to “closed dark umbrella (Figure 1), then, a second…”

Line 106-7: reformulate..”The horses were released, the umbrella was suddenly opened, and T3 sample (stress time) was obtained.”

Line 109: change to “Finally, once the horses were in their box stalls, two additional saliva samples were taken 30 min (T4) and 60 min (T5) after T3”.

Line 110-111: change to “were tested on the same day from 9:00 to 12.30 a.m.”

Line 112: change to “horses could not see what happened in the inside of the arena”

Line 123: add a comma “trot or gallop (e), passage”

Line 120: why do only some behavioural patterns have references?!

Line 138: I would recommend to indicate the meaning of the abbreviations in each section (intro, M&M, results, etc.) otherwise an abbreviation list should be provided at the beginning.

Line 152: change to “Previously validated in saliva for horses [30] and pigs [32].”

Line 162: please indicate what the abbreviations of the index of laterality stand for…

Figure 2: Please write the analyte represented on each graph on the y-axis

Discussion

I would recommend to indicate the meaning of the abbreviations in each section (intro, M&M, results, etc.) otherwise an abbreviation list should be provided at the beginning.

Line 237: please change to “sheep [20] and pigs [21,3]”

Line 241: please change to “This increase occurs sooner than..”

Line 242: please change to “salivary cortisol in horses increases only 30 min after ACTH”

Line 263: a p value of 0.057 means the difference seen is NOT significant! I would not accept to use “tendency” or “trend” in a scientific manuscript when referring to something that is “almost significant”, I believe there is only “significant” and “not significant”

Line 269: Please change to “Although ADA did not change significantly after stress, it correlated negatively with the index of laterality, with the preferential use of the left eye, and positively with BChE”

Line 281: change to “sex”

Line 285: change to “collection would have been”

Reviewer 2 Report

This is an interesting paper describing preliminary results of a correlation between saliva biomarkers and behavioural reactions to stress in horses. The manuscript is well written and the methods and results are well described and discussed. Even there are some limitations of the study regarding a small number of animals of different breeds, the manuscript could be a good starting point for further investigation of salivary biomarkers for evaluation of stress in horses.

There is a minor point that the authors should address:

Figure 2: Each graph should contain the name of the parameter measured on the y-axis to be more clear and self-explaining.

Reviewer 3 Report

This is a well written manuscript and the results, although preliminary, provide surprising and new information on the effect of an umbrella test on some saliva analytes rarely measured in horses and their relationship with behavior. Surprising because of the pronounced reaction of most analytes measured to such a short and kind of mild stress like the umbrella test. The manuscript would benefit of a detailed discussion by the authors of the following details:

-the statement “emotionality in horses is important as it may increase the risk of equestrian accidents”. Does it have fundament?

-the temporal dynamic of behavioral parameters. Has this been examined for any of these? If not: would this not make sense to be done too, to provide information about the stress response? If yes: Then the statement is wrong and the salivary or any other analytes would provide complementary information only.

-the emotionality index. It seems to be a rather arbitrary index. Has it been “validated” somehow? What happens to the relationships with the saliva analytes when the order of the “values” given to the behavioral parameters is changed and the index calculated again?

Specific comments

Write behavior always like American English or behaviour as British English.

Page 2

line 59 delete “sampling”.

Line 71-73 is written: In addition, other salivary analytes such as total esterase (TEA), butyrylcholinesterase (BChE) and lipase may increase at different levels in pigs and sheep after acute stress situations.”

Why it is written that they “may” increase? The graphs show that they increase, some more some less, but they increase. Or is here a misunderstanding?

Line 89 delete “housing”

Page 3

Line 110 replace “a” for “the”

Page 4

Line 177 add “there was a” after “whether”.

Page 9

Line 263 delete “d” from “closed”

Line 279 delete “s” from “kinds”

Line 283 replace “the” for “to”

Line 285 replace “c” in “could” for “w”

Further

Do not mention the 10th This info does not provide anything to the manuscript. Explain in Materials and Methods when exactly the T3 saliva sample was collected? It was not in the first minute after the stress because then the behavior of the horses was filmed. Immediately thereafter, or after bringing the horse into the box? How much time elapsed between stress and sampling? In the legend of figure 2 replace the “b” after cortisol for “f” Table 1 would benefit from

-write “horse” above the numbers 1 to 9

-a correction or explanation because in the first column “Behavioral or Behavioural patterns, frequencies of occurrence” % is the unit. % of what? Are the numbers depicted not simply counts?

Figure 3: Why are these 7 graphs shown only?
